# Fast and High-Quality Monocular Depth Estimation with Optical Flow for Autonomous Drones

**Tomoyasu Shimada [1], Hiroki Nishikawa [2] , Xiangbo Kong [1] and Hiroyuki Tomiyama [1,*]**

[1] Graduate School of Science and Engineering, Ritsumeikan University, Kusatsu 525-8577, Japan
[2] Graduate School of Information Science and Technology, Osaka University, Suita 565-0871, Japan
[*] Correspondence: ht@fc.ritsumei.ac.jp; Tel.: +81-77-561-4928

**Abstract:** Recent years, autonomous drones have attracted attention in many fields due to their convenience. Autonomous drones require precise depth information so as to avoid collision to fly fast and both of RGB image and LiDAR point cloud are often employed in applications based on Convolutional Neural Networks (CNNs) to estimate the distance to obstacles. Such applications are implemented onboard embedded systems. In order to precisely estimate the depth, such CNN models are in general so complex to extract many features that the computational complexity increases, requiring long inference time. In order to solve the issue, we employ optical flow to aid in-depth estimation. In addition, we propose a new attention structure that makes maximum use of optical flow without complicating the network. Furthermore, we achieve improved performance without modifying the depth estimator by adding a perceptual discriminator in training. The proposed model is evaluated through accuracy, error, and inference time on the KITTI dataset. In the experiments, we have demonstrated the proposed method achieves better performance by up to 34% accuracy, 55% error reduction and 66% faster inference time on Jetson nano compared to previous methods. The proposed method is also evaluated through a collision avoidance in simulated drone flight and achieves the lowest collision rate of all estimation methods. These experimental results show the potential of proposed method to be used in real-world autonomous drone flight applications.

**Keywords:** depth estimation; optical flow attention; perceptual discriminator

## 1. Introduction

There has been growing demand for Unmanned Aerial Vehicles (UAVs), so called drones, with diverse capabilities in many fields such as agriculture, surveillance, logistics, military, and so on [1]. The term of the drone is commonly known as remote (or autonomous) flying robots, but in fact, this term is also used to describe a variety of vehicles such as submarines or land-ropers. Focusing on drones as flying robots, commonly known as drones, we classified them into three types derived from the flying mechanisms [2]: Multi-rotor drones, fixed-wing drones, and hybrid-wing drones. Multi-rotor drones have a set of rotary wings and are based on Vertical Take-Off and Land (VTOL) principle, and then they need to tilt for producing driving horizontal force. Fixed-wing drones can be imagined such as small airplanes, which have a higher potential of glide to fly fast than multi-rotor ones [3]. Unlike multi-rotor drones, fixed-wing drones require a runway to take off and land due to their Horizontal Take-Off and Landing (HTOL) nature. As the other type of drones, hybrid-wing drones are utilized. They are called hybrid with regard to including fixed and rotary wings and can flexibly take advantage of either wing depending on the flight situation.

These drones are adopted depending on their usage. Especially for military usage, fixed-wing drones could be employed since the military has a runway long enough for such drones to be ready for taking off and landing [4]; however, commercial use of drones such as logistics and smart city surveillance can hardly prepare the specialized runway so that multi-rotor drones have been commonly employed in the literature so far [5].

In order to realize fully autonomous flight of drones, there must be an essential function that enables drones to perceive their surroundings and avoid collision with obstacles [6]. There have been many works on obstacle avoidance for several decades and they have adopted depth perception methods through distance sensors such as ultrasonic sensors, LiDAR, Microsoft Kinect, or stereo cameras [7–10]. For instance, the authors employed high-performance sensors such as ultrasonic sensors and radars in the proposals [11–13]. These methods achieved collision avoidance by acquiring the distance between the drone and the obstacle from the sensor and using an obstacle-free location as a waypoint. However, such sensors have a limitation of either range of distance or energy consumption due to battery capacity on drones and are not suitable for autonomous drones. On the other hand, several works employ LiDAR as a long-distance measurable sensor [7,8]. Although these methods have realized high-speed autonomous flight, the flight distance of a drone equipped with LiDAR is limited due to the high energy consumption caused by its weight. For example, the weight of HDL-64E is well-known as 12,700 g [14]. Therefore, to make effective use of drones, it is necessary to estimate the long distance from light sensors. Instead of such sensors, a monocular camera as the depth sensor for obstacle recognition and avoidance has been an attractive solution for autonomous drones due to several advantages of the monocular camera, including ease of use, lightweight, small footprint, and low power consumption. Although in order to fly safely, vision-based methods require Deep Neural Networks (DNNs)-based algorithms with high power consumption, with the advancement of embedded systems technology, it is now possible to use small form-factor, low-power-consumption board computers (e.g., Jetson Nano and Jetson Xavier NX) that can be installed on drones for DNN-driven computations.

Most monocular depth estimation methods usually employ Convolutional Neural Networks (CNNs). In addition, high-accuracy depth estimation networks consist of large-scale CNN in many cases [15–18]. Several methods have achieved practical accuracy in long-depth estimation from a monocular camera using CNN. In the studies [19,20], the authors employed the two steps CNN to estimate depth accurately. In the first step, the network takes a monocular image and outputs a global rough depth map. The network refines the depth map locally from a monocular image and the global rough depth map in the second step, but the drawback of this network is low accuracy for the network size. To solve this problem, various efforts have been made to improve the accuracy of depth estimation.

One way to improve accuracy is to deepen and expand the network. In the work [17], the authors employed Visual Geometry Group (VGG) to extract features from a monocular image for depth estimation. The architecture of VGG is characterized by its simple design and the use of multiple small convolutional filters, which led to its excellent performance on large-scale image classification benchmarks [21] and the paper [17] achieved higher accuracy than the works in [19,20]. These methods [15,16,18,22,23] also used other deep CNNs (e.g., ResNet50 [24] and DenseNet121 [25]). In addition, the authors added the transformer techniques to the CNN depth estimator to refine the depth estimation map [26,27]. The deep CNN [15,16,22] or transformer [26–28] methods improve accuracy significantly. However, the inference time of these methods is too long for drones to fly safely, especially on a low or middle-grade Graphical Processing Unit (GPU), which can be loaded on drones. In order to utilize high accuracy and large-scale deep CNN-based methods, drones communicating with cloud servers have been recognized as an efficient solution in that the computation is conducted very quickly on a cloud server instead of on a drone itself. However, the latency between a drone and a cloud server usually takes up to a few seconds and it also jeopardized to security vulnerabilities, resulting in fatal systematical problems [2,29–31].

Another way to improve the depth estimation method is to pre-process a monocular image and use lightweight CNN. For instance, the authors in [32] employed semantic segmentation as a pre-processing to improve accuracy. This method [32] identifies categories and calculates instance segmentation maps followed by dividing them into patches from

categories and instance maps. In the next step, this method estimates the depth of each category and each instance. As a result, by category and instance-wise depth estimation, the computational cost for depth estimation is small, and the edges are sharper and more accurate than the other methods. However, the authors employed ResNet50 for segmentation, which increases the computational cost for pre-processing. Therefore, pre-processing for depth estimation on drones must process fast not to increase the total inference time of depth estimation.

The authors in [33,34] employed Pix2Pix [35] for depth estimation and demonstrate that inference time is practical for autonomous flight. The authors [34] concluded the estimation accuracy of this method [33] is not enough for collision avoidance completely; therefore, the authors in [34] employed optical flow as a pre-processing and improved accuracy. This method [34] generates optical flow maps and replaces a part of monocular image pixels with the optical flow map pixels. However, this method [34] does not utilize full optical flow information. The work in [36] proposed a method that inputs two images into CNN and this method worked well to use all information in the image. The method [36] employed ORB-SLAM presented in [37], which is known as a Simultaneous Localization and Mapping (SLAM), as a pre-processing and generated a sparse depth map. SLAM is a technology that allows a device to create a map of its surroundings and determine its location within that map in real time without Global Positioning Systems (GPS). SLAM algorithms are computationally expensive, and real-time performance can be difficult to achieve on limited computational resources. In addition, the simplified depth map generated by the SLAM algorithm is a sparse depth map like a LiDAR cloud point map. However, due to the characteristics of CNNs, it is difficult to extract point cloud features from sparse depth maps. Stereo photogrammetry, Structure from Motion (SfM) photogrammetry and Multi-View Stereo (MVS) are also a process of creating 3D images from 2D images by using the stereogram principle, which involves the use of stereo imagery to derive 3D information about a scene. However, these methods need high-resolution cameras to create accurate 3D models. Furthermore, these methods can only obtain a 3D model if multiple images are acquired. Therefore, to use these methods for drone collision avoidance, it is necessary to pre-flight the flight route and create a 3D map in advance, which is not suitable for dynamic environments.

There exist some works that employ Conditional Generative Adversarial Networks (CGAN) [38] to train CNN for depth estimation for its easy adaptation for many image-to-image problems [33,34,39,40]. CGAN improves the performance of the generator by feeding additional data to the generator and the discriminator. However, it is not enough to determine true/false with only one discriminator, and it is effective to determine true/false from other perspectives.

In this paper, we propose fast and effective pre-processing for depth estimation and CNN to utilize pre-processing information. To further enhance our proposal, we adopt a perceptual discriminator to improve the accuracy of depth estimation without increasing the complexity of the generator network.

The rest of this paper is organized as follows. Section 2 describes a proposed method for depth estimation using optical flow attention and a perceptual discriminator. The experiments on the evaluation of error, accuracy, and collision rate are presented in Section 3, and Section 4 shows the results of these experiments. Section 5 describes discussions about experimental results and issues of our proposed method. Section 6 concludes this paper.

## 2. Proposed Method

This section describes a proposed depth estimation method with optical flow attention. Figure 1 shows system overview of the proposed method.

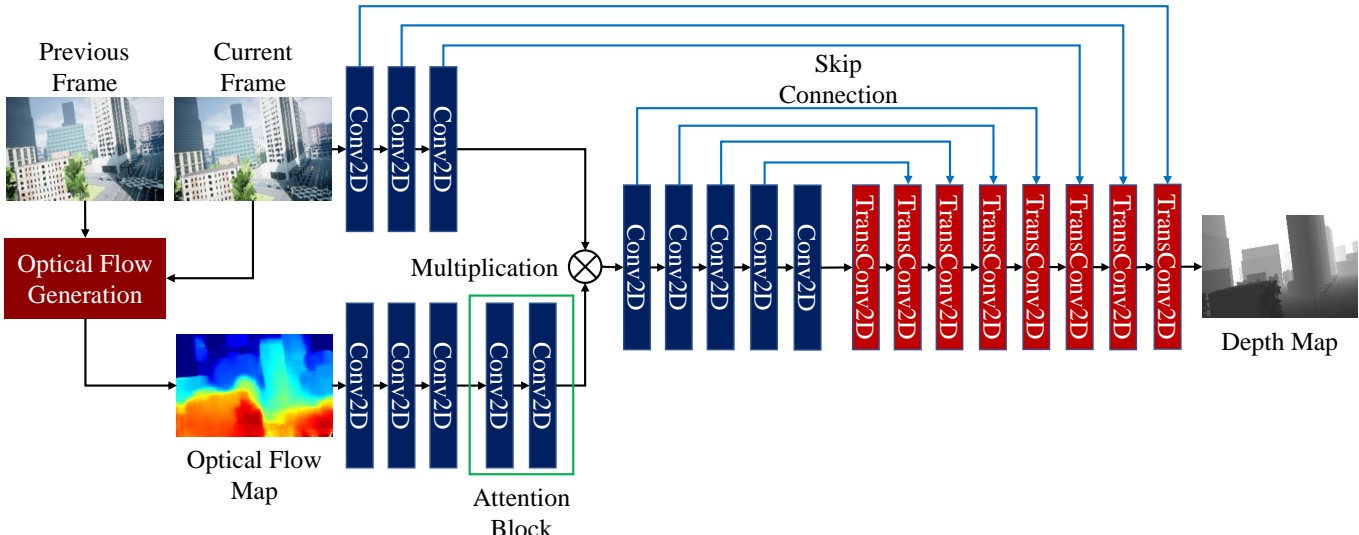

**Figure 1.** The proposed method.

As shown in Figure 1, the proposed method has two main processes. Firstly, the proposed method generates optical flow from serial frames. Secondly, the proposed method estimates depth using CNN with attention, which takes a monocular image and an optical flow map. In addition, a discriminator different from the conventional CGAN is added during the training of the estimator to improve accuracy. In Setion 2.1, we introduce optical attention and depth estimation in detail. In Setion 2.2, we detailed present the perceptual discriminator used in training.

### 2.1. Optical Flow Attention and Skip Connection

We focused on the fact that drones are a form of mobility, and the continuous frames acquired by the camera on a moving drone show small movements of distant objects and large movements of nearby objects. The optical flow can obtain the movement vectors of objects in the image, and in drones, the displacement of the optical flow can be regarded as a simplified depth. We employ the Farnebäck method [41] to generate optical flow since this method can obtain dense optical flow. The Farnebäck method [41] approximates pixel luminance values with a second-order polynomial and compares the coefficients between frames to estimate the amount of movement with high accuracy. Let $f_t(x) \in [0, 1]$ denote the luminance value of coordinate $x$ at time $t$. The luminance values in the neighborhood of $x$ are expressed as second-order polynomials, and the coefficients are optimized by the weighted least-squares method in Equation (1).

$$\hat{f}_t(x) = x^T A_t x + b_t^T x + c_t \tag{1}$$

$A_t$, $b_t$, $c_t$ are a (2,2) symmetric matrix, a (2,1) column vector, and a scalar, respectively. The amount of movement $v_t$ at time $t$ can be estimated from Equation (2). For details refer to [41].

$$\hat{f}_t(x) = \hat{f}_{t+1}(x + v_t) \tag{2}$$

In Figure 1, this optical flow map is regarded as simplified depth. However, this map is noisy, and when this map is input CNN for depth estimation, the output of depth estimation affected by noise is low accuracy. Therefore, we propose a network using attention to fully utilize this simplified depth image and minimize the effect of noise. As shown in Figure 1, this network has thirteen convolutional layers and eight transposed convolutional layers. The kernel size is four, and the stride is two in these layers. In addition, between these layers, there is ReLU as an activation function and batch normal-

ization. In the attention block, the features of the optical flow map extracted by the two convolutional layers are normalized from 0.0 to 1.0 by the Sigmoid function, and the output channel size is the same as the monocular image features before multiplication to determine the attention area in each channel. Finally, the output of the attention block is multiplied by the feature values of the monocular image. In past work, some pixels in the optical flow map are selected at certain intervals and added to the RGB map [34]. This approach does not guarantee that the optical flow pixels containing key information are selected, and at the same time, it is possible to select noise points and add them to the RGB image. In this work, we select the pixels to be added through the attention block. The attention block can determine which areas and channels are not needed from the information in the optical flow map when the output of the attention block is close to 0, and which areas and channels should be focused on in the next layer when the output of the attention block is close to 1. Since noisy optical flow features are used for the attention block, the attention block can reduce the impact on the output of depth estimation and maximize the use of the optical flow map.

In addition, we employ skip connections inspired by U-Net [42]. The deeper the convolutional layer, the more local features can be extracted, but location information becomes ambiguous. The skip connection gives the decoder outputs of the encoder at the same depth, and it is possible to recover positional information and extract local features. However, when a skip connection is connected to the optical flow, the depth estimation results are less accurate due to the noise of the optical flow. Therefore, the skip connections are only connected to monocular image layers.

### 2.2. Perceptual Discriminator

We train the estimator based on the Pix2Pix training method [35]. However, when the estimator is trained using the Pix2Pix training method as it is, the accuracy is not improved enough. Therefore, we add a perceptual discriminator to improve the accuracy of depth estimation inspired by [43]. Figure 2 shows the overview of training.

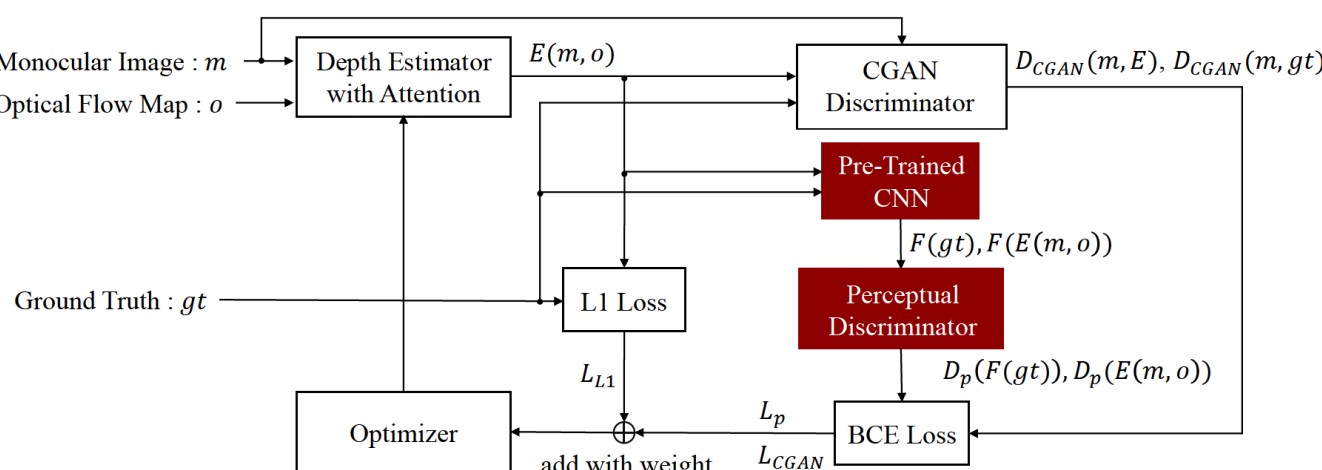

**Figure 2.** Overview of training

In Figure 2, there are white blocks and red blocks. The white blocks are the Pix2Pix training method, and the red blocks are our added training method. As shown in Figure 2, the perceptual discriminator takes features extracted from pre-trained CNN. In training, the parameters of pre-trained CNN are frozen, and the perceptual discriminator identifies whether the input is the features of ground truth or depth estimation. Therefore, by employing a perceptual discriminator, the depth estimator can make the estimation results

resemble ground truth in terms of features. The loss function of the perceptual discriminator shows in Equation (3).

$$\mathcal{L}_p(E, D_p) = \mathbb{E}_{gt}[\log D_p(F(gt))] + \mathbb{E}_{m,o}[\log(1 - D_p(F(E(m,o))))] \tag{3}$$

Here, $m$ is a monocular image, $gt$ is a ground truth of the depth map, $E$ is an output of the estimator, $F$ is features extracted by pre-trained CNN and $D_p$ is an output of the perceptual discriminator.

$$\mathcal{L}_{CGAN}(E, D_{CGAN}) = \mathbb{E}_{m,gt}[\log D_{CGAN}(gt, m)] + \mathbb{E}_{m,o}[\log(1 - D_{CGAN}(E(m,o), m))] \tag{4}$$

Here, $D_{CGAN}$ is an output of the CGAN discriminator. An additional intelligence $m$ is added to the input to improve performance compared to conventional GAN. $\mathcal{L}_p$ and $\mathcal{L}_{CGAN}$ are predicated on binary logarithmic loss, and $D_p$ and $D_{CGAN}$ are probabilities which these discriminators identify an input as ground truth. Also, these losses only add more relevance to the estimator, and the error is likely to be large for depth estimation. Therefore, we use the L1 loss as in Pix2Pix to give an absolute value error as shown in Equation (5).

$$\mathcal{L}_{L1}(E) = \mathbb{E}_{m,o,gt}[||gt - G(m,o)||_1] \tag{5}$$

Finally, we define the total loss as:

$$\mathcal{L}_{total} = \alpha \mathcal{L}_{CGAN} + \beta \mathcal{L}_p + \gamma \mathcal{L}_{L1} \tag{6}$$

Here, $\alpha$, $\beta$, and $\gamma$ are weights of each loss function.

## 3. Experiments

In this section, we evaluate our method in terms of accuracy, latency, and performance to avoid collisions. The dataset, which is used for training, validation, and testing, has been collected from the four maps provided in the AirSim environment; Blocks, City, Coastline, and Neighborhood, where the overview of the maps is shown in Figure 3.

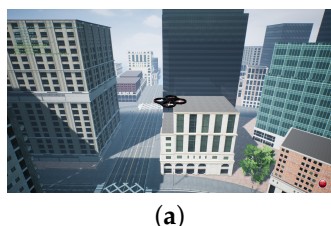 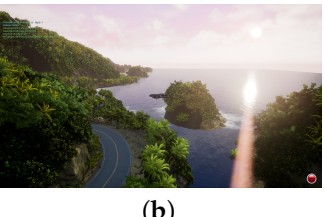 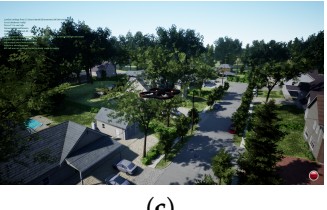 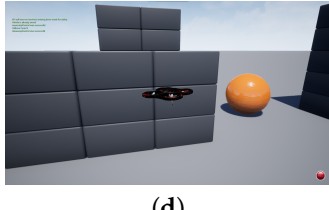

(**a**)　　　　　　　　(**b**)　　　　　　　　(**c**)　　　　　　　　(**d**)

**Figure 3.** Appearance of the maps for training: (**a**) City environment, (**b**) Coastline, (**c**) Neighborhood, (**d**) Blocks.

AirSim [44] is a kind of flight simulator that uses a virtual environment called Unreal Engine 4. This simulator faithfully reproduces reality in visual information and physics.

### 3.1. Implementation

In AirSim [44], we can obtain depth maps and monocular images in the same frame so that we create a dataset that is suitable for autonomous drones. We have obtained the data through the flight simulation with a quadcopter built-in AirSim, and the dataset has 9000 pairs as training data, 1000 pairs as validation data, and 1000 pairs as test data.

We used Intel Core i9-10900K (32 GB of main memory) and NVIDIA GeForce RTX 3090 in training our models. We train our model in the following conditions we manually optimized these parameters: the number of epochs is set to 500. The batch size is set to 32, $\alpha$ is 0.5, $\beta$ is 0.5, and $\gamma$ is 100.

In addition, in order to compare other depth estimation methods not using AirSim dataset, we train our model using KITTI dataset [45]. KITTI dataset [45] is a famous dataset to evaluate depth estimation and has real-world depth maps and monocular images. KITTI with Eigen split has 22,600 pairs as training data, 888 as validation data, and 697 pairs as test data. In training our models using the KITTI dataset, we employ Eigen's split method and the following conditions and we manually optimized these parameters: the number of epochs is set to 200. The batch size is set to 8, $\alpha$ is 0.5, $\beta$ is 0.5, and $\gamma$ is 100. In addition, we employ fine-tuning from the pre-trained AirSim dataset model in training since our model needs more effective training because of a small network. The epoch number is determined based on the convergence of the loss function. The parameters $\alpha$, $\beta$ and $\gamma$ of the loss function are set to 0.5 because $\alpha$ and $\beta$ are the losses associated with the discriminator, respectively, and both need to give the same ratio of losses. $\gamma$ is determined with reference to Pix2Pix [35]. These were determined using the same values for both the AirSim and KITTI datasets. The batch size was determined based on the available GPU memory of the computer used for training.

In addition, The models used in the comparison, CycleGAN [46], Shimada et al. [33] and Shimada et al. [34] are trained by ourselves on the AirSim dataset. In Shimada et al. [33] and Shimada et al. [34], the parameters are the same as in these papers. The parameters of CycleGAN [46] is the weight of identity loss is 5, the weight of cycle consistent loss is 10, the number of epochs is 200, and the batch size is 16 based on CycleGAN paper [46]. The models on the KITTI dataset are compared using the results in these papers. In addition, the inference time evaluations use publicly available neural networks.

### 3.2. Accuracy, Error and Latency Evaluation and Ablation Study

In order to quantify the estimation error of models, we use root mean squared error (RMSE) and absolute relative error (Rel.) metrics. Hereby, RMSE is obtained by the following equation.

$$RMSE = \sqrt{\frac{1}{N}\sum_{i=1}^{N}(y_i^{gt} - y_i)^2} \tag{7}$$

$y_i^{gt}$ is ground truth value. $y_i$ is estimation value. $N$ is a number of data. Rel. is obtained by the following equation.

$$Rel. = \frac{1}{N}\sum_{i=1}^{N}\frac{||y_i^{gt} - y_i||}{y_i^{gt}} \tag{8}$$

Specifically, the accuracy metrics are defined as:

$$\delta_n = \frac{\left\{x \mid y_i : max\left(\frac{y_i}{y_i^{gt}}, \frac{y_i^{gt}}{y_i}\right) < 1.25^n\right\}}{N} \qquad (n = 1, 2, 3), \quad (i \in (1, ..., N)) \tag{9}$$

In addition, we evaluate inference time in order to analyze the inference time. In this experiment, we compare our proposed method with the 6 methods which are Eigen et al. [19] Shimada et al. [33], Shimada et al. [34], CycleGAN [46], Xin et al. [36], and Kuznietsov et al. [18], respectively. To compare these methods, we reproduce the networks from published papers and measure inference times. Also, we use the three devices which are Jetson Nano, Jetson Xavier NX, and GeForce RTX 2070 SUPER, respectively. The network inputs are the images which are $256 \times 256$ resolution and the channels are three based on the AirSim dataset. In KITTI dataset, the network inputs are the images which are resized in height is 256, and width is 1024 to be able to compute Jetson Nano. This experiment is performed 1000 times consecutively for each method, and the average inference time is evaluated.

### 3.3. Collision Rate Evaluation

Previously, we have evaluated the accuracy and runtime of the proposed method. In this section, we conduct the simulation of a drone flight in AirSim to demonstrate that the proposed method can fly avoiding collision with objects. In order to realize the safe flight of an autonomous drone, it is necessary to plan the path by itself, that is, the drone needs to select the direction so that the drone can avoid colliding with objects. In the experiments, we use a state-of-the-art path planning method for flight control, which is developed in [47]. The work in [47] introduces a method that divides a depth map into multiple sections. The presented method in [47] divides a depth image into 289 overlapped sections (17 rows and 17 columns) as shown in Figure 4.

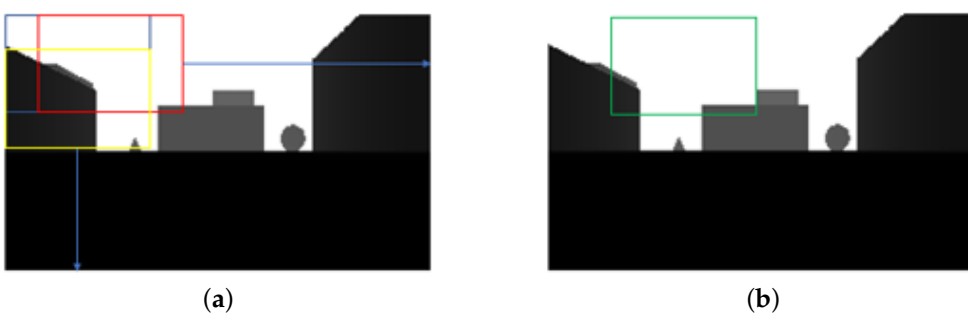

| (a) | (b) |

**Figure 4.** Direction decision from divided sections in [47]. (**a**) Overlapped section, (**b**) Section selection.

By dividing into overlapped sections, the drone selects the best section to avoid obstacles and pass safely so that the drone determines the section with the maximum total pixel value. The flight is simulated 400 times in the four environments. The flight scenarios are randomly generated in terms of route, direction, and distance. We compare the collision rates that the number of collisions accounts for towards the total number of flights. Hereby, we define the collision rate for a map in the following equation:

$$Collision\ Rate = \frac{No.\ of\ Collisions}{No.\ of\ Flights\ (i.e.,\ 400\ flights\ in\ total)} \quad (10)$$

Note that we assume that the flight has a collision if the drone collides with an obstacle even once during its flight.

In the experiments, we use the following four methods: The first can measure up to 10 m, which assumes a real depth camera for a reasonable price and is light-weight enough to be equipped on a drone. The second can measure up to 255 m. This method assumes an ideal depth camera, where it can measure up to 255 m but is too heavy to be mounted on a drone in the real world. This method is used as ground truth depth images for comparison. The third and the fourth are presented by Shimada et al. [33] and [34]. This method inputs a monocular image to generate a depth image through Pix2Pix. The fifth is our proposed method. In this experiment, we use Intel Core i7-9700K (64 GB of main memory) and NVIDIA GeForce RTX 2070 SUPER. In addition, the depth estimation is performed on the CPU to avoid the discrepancy between the drone with a low-grade GPU and the inference time. Our method combines an image with an optical flow map into Pix2Pix, and it generates the estimated depth map.

## 4. Results

### 4.1. Accuracy, Error and Latency Evaluation

Table 1 shows the results of error and accuracy using the AirSim dataset, and Figure 5 shows the inputs and outputs of Shimada et al. method [34] and ours.

**Table 1.** Error and accuracy evaluation using AirSim dataset.

| | Error (↓) | | Accuracy (↑) | | |
|---|---|---|---|---|---|
| | RMSE | Rel. | $\delta_1$ | $\delta_2$ | $\delta_3$ |
| Shimada et al. [33] | 5.924 | 0.133 | 0.882 | 0.956 | 0.977 |
| Shimada et al. [34] | 5.917 | 0.131 | 0.886 | 0.957 | 0.982 |
| Cycle GAN [46] | 5.961 | 0.135 | 0.891 | 0.960 | 0.984 |
| Ours | **5.771** | **0.131** | **0.898** | **0.970** | **0.994** |

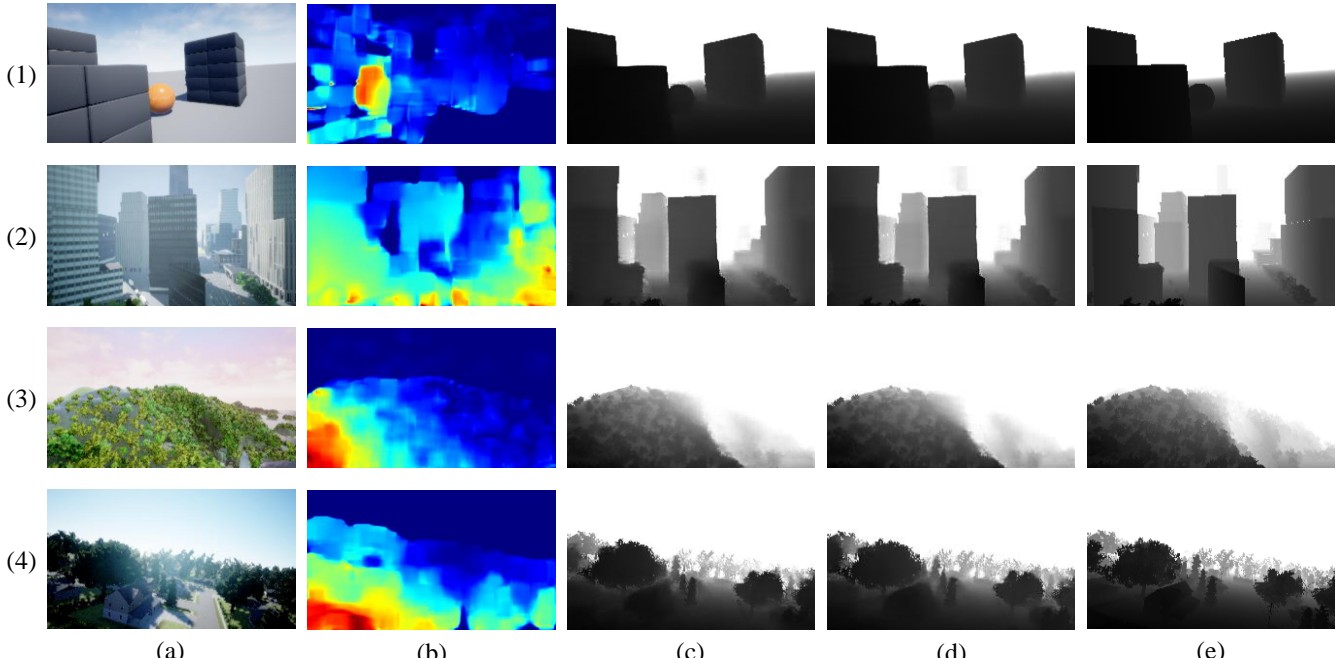

**Figure 5.** Inputs and outputs each method: (**a**) monocular image, (**b**) optical flow map, (**c**) Shimada et al. [34], (**d**) ours, (**e**) ground truth, (1) Blocks, (2) City Environment, (3) Coastline, (4) Neighborhood.

As shown in Table 1, our proposed method is superior to the other methods. In Shimada et al. method [33] and Cycle GAN method [46], these networks take only a monocular image, and these methods cause low accuracy because of limitations of extracted information. In Shimada et al. method [34], this method generates optical flow and a part of pixels in a monocular image replaced to pixels of optical flow maps so that optical flow information is misses when replacing pixels, causing low accuracy.

As shown in Figure 5, in the block, the output of Shimada et al. method [34] is blurred around the orange sphere. In addition, some white dots are visible in other outputs. We have identified an outlier due to the influence of the replaced pixels in the optical flow. On the other hand, the proposed method has no such outliers and almost no blurring of edges such as spheres. The rational use of optical flow based on the attention method is considered to be the reason why the proposed method is superior to other methods.

Table 2 shows the evaluation accuracy and error using KITTI dataset [45].

**Table 2.** Error and accuracy evaluation using KITTI dataset with Eigen's split.

| | Error (↓) | | Accuracy (↑) | | |
|---|---|---|---|---|---|
| | **RMSE** | **Rel.** | $\delta_1$ | $\delta_2$ | $\delta_3$ |
| Eigen et al. [19] | 7.156 | 1.515 | 0.692 | 0.899 | 0.967 |
| Liu et al. [23] | 6.986 | 0.217 | 0.647 | 0.882 | 0.961 |
| Kuznietsov et al. [18] | **4.621** | **0.113** | 0.862 | 0.960 | 0.986 |
| Shimada et al. [34] | 7.605 | 0.154 | 0.813 | 0.958 | 0.985 |
| Xin et al. [36] | 5.752 | 0.125 | 0.869 | 0.956 | 0.980 |
| Ours | 4.712 | 0.121 | **0.870** | **0.973** | **0.992** |

As shown in Table 2, Kuznietsov et al. [18] achieves the lowest errors in RMSE and Rel. and our proposed method is the highest accuracy in all $\delta$. In comparison to these methods, the accuracy of our proposed method on $\delta_1$, $\delta_2$, and $\delta_3$ respectively is 7.0 percent, 1.6 percent, and 0.7 percent better than Shimada et al. [34] since the proposed method can utilize optical flow information by using attention block more than Shimada et al. [34]. In Kuznietsov et al. [18] method, the network uses ResNet 50 as an encoder. Although the network is deeper by far than our proposed method, and the estimation results have a larger percentage of zero errors and lower RMSE and Rel. than our proposed method, running a deeper network generally consumes more power. The high energy consumption is not good for drones in terms of their battery capacity. On the other hand, in $\delta$, our proposed method is superior to Kuznietsov et al. [18] since the estimation results of our proposed method have a small number of outliers by taking an optical flow map as a simplified depth map.

Tables 3 and 4 shows the results of inference time evaluation.

**Table 3.** Inference time evaluation using AirSim dataset.

| Method | Inference Time [ms] | | |
|---|---|---|---|
| | **Nano** | **Xavier** | **RTX 2070 SUPER** |
| Eigne et al. [19] | 8.8 | 5.1 | 1.4 |
| Shimada et al. [33] | 18.4 | 13.4 | 2.9 |
| Shimada et al. [34] | 18.0 | 12.5 | 2.7 |
| CycleGAN [46] | 32.1 | 23.1 | 4.5 |
| Xin et al. [36] | 33.4 | 22.2 | 4.8 |
| Kuznietsov et al. [18] | 70.1 | 39.1 | 9.6 |
| Ours | 23.8 | 14.5 | 3.6 |

**Table 4.** Inference time evaluation using KITTI dataset.

| Method | Inference Time [ms] | | |
|---|---|---|---|
| | **Nano** | **Xavier** | **RTX 2070 SUPER** |
| Eigne et al. [19] | 9.5 | 7.0 | 1.6 |
| Shimada et al. [33] | 18.4 | 12.9 | 3.1 |
| Shimada et al. [34] | 17.7 | 13.4 | 3.0 |
| CycleGAN [46] | 34.5 | 24.6 | 5.7 |
| Xin et al. [36] | 35.2 | 24.8 | 5.5 |
| Kuznietsov et al. [18] | 76.3 | 41.3 | 12.0 |
| Ours | 25.9 | 16.4 | 4.3 |

As shown in Tables 3 and 4, Eigen et al. [19] achieves the shortest inference time on all devices, and the proposed method has the second shortest inference time. Shimada et al. [34] and Shimada et al. [33] use Pix2Pix, and our network adds an attention technique to Pix2Pix, and the inference time is increased only by the computational load. On the other

hand, the inference time is 2.95X, 2.70X, and 2.67X shorter on each device than method Kuznietsov et al. [18], whose accuracy is competitive with the proposed method. Therefore, considering the accuracy and inference time, the proposed method is the most suitable method for autonomous drones.

### 4.2. Ablation Study

A total of eight combinations is used to evaluate error and accuracy as an ablation study. Table 5 shows the results of the ablation study using the AirSim dataset.

**Table 5.** Ablation study using AirSim dataset.

| Perceptual Discriminator | Optical Flow Attention | Skip Connection | Error (↓) | | Accuracy (↑) | | |
|:---:|:---:|:---:|:---:|:---:|:---:|:---:|:---:|
| | | | **RMSE** | **Rel.** | $\delta_1$ | $\delta_2$ | $\delta_3$ |
| | | | 6.721 | 0.244 | 0.774 | 0.887 | 0.928 |
| ✓ | | | 6.227 | 0.237 | 0.768 | 0.883 | 0.923 |
| | ✓ | | 6.290 | 0.226 | 0.771 | 0.888 | 0.929 |
| | | ✓ | 5.942 | 0.134 | 0.887 | 0.956 | 0.977 |
| ✓ | ✓ | | 6.287 | 0.236 | 0.768 | 0.887 | 0.930 |
| ✓ | | ✓ | 5.951 | 0.206 | 0.818 | 0.915 | 0.950 |
| | ✓ | ✓ | 5.929 | 0.131 | 0.887 | 0.960 | 0.986 |
| ✓ | ✓ | ✓ | **5.771** | **0.131** | **0.898** | **0.970** | **0.994** |

As shown in Table 5, the combination of all the factors is the best result and combinations without skip connections have significantly worse accuracy and error. In addition, the combinations that include the optical flow attention have a higher accuracy than the combinations without the optical flow attention, and the combinations involving the perceptual discriminator have greatly improved accuracy for combinations with a certain level of accuracy. Figure 6 shows the outputs of each combination.

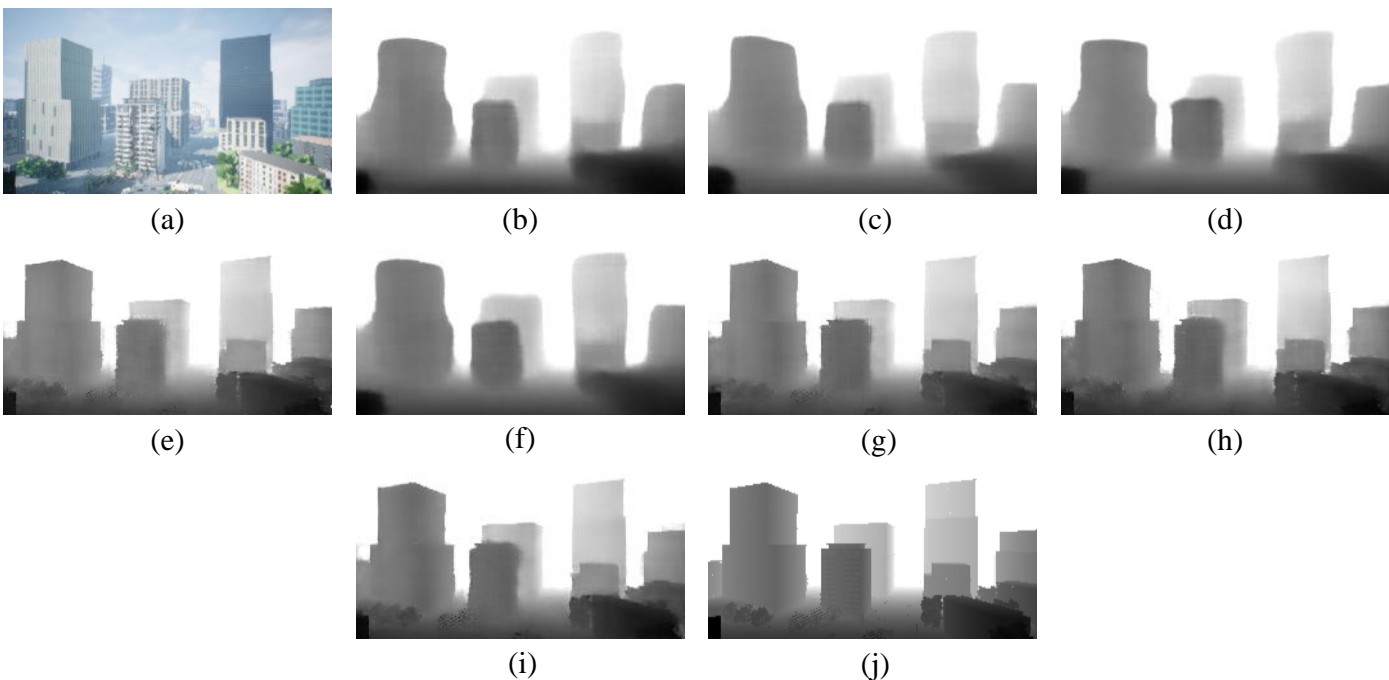

**Figure 6.** Ablation study: (**a**) monocular image, (**b**) nothing, (**c**) perceptual discriminator, (**d**) optical flow attention, (**e**) skip connection, (**f**) perceptual discriminator and optical flow attention, (**g**) perceptual discriminator and skip connection, (**h**) optical flow attention and skip connection, (**i**) ours, (**j**) ground truth.

As shown in Figure 6b–d,f, the outputs of the combinations without the skip connection have fuzzy edges and the overall image appears blurred. This fuzzy edge causes significantly decreased accuracy. Therefore, skip connections can recover location information that becomes ambiguous when the network extracts local features. As shown in Figure 6d,f,h,i, the outputs of the combinations with the optical flow attention have a more accurate depth than without it. Therefore, the results show that optical flow information is correlated with depth, and optical flow attention is an effective use of this information. As shown in Figure 6c,f,g,i, the outputs of the combinations including the perceptual discriminator have been able to estimate the depth of thin objects such as trees, which are lost in the other combinations. These results demonstrate the perceptual discriminator can distinguish vanishing objects from the input features and propagate them back to the estimator as losses. Therefore, all of the proposed factors are essential for a lightweight and accurate depth estimation network.

### 4.3. Collision Rate Evaluation

Table 6 shows the results of collision rate evaluation.

**Table 6.** Comparison of collision rate.

| Map | Collision Rate (%) | | | | | |
|---|---|---|---|---|---|---|
| | 10 m | 255 m | Shimada [33] | Shimada [34] | CycleGAN [46] | Ours |
| Blocks | 58.75 | 7.000 | 17.50 | 14.50 | 15.00 | **5.50** |
| City environment | 73.50 | **26.00** | 34.75 | 34.00 | 36.50 | 26.50 |
| Coastline | 70.50 | 0.250 | 1.500 | 1.250 | 1.500 | **0.00** |
| Neighborhood | 82.00 | 7.000 | 2.500 | 1.000 | 5.500 | **0.25** |

As shown in Table 6, in City Environment, the method using ground truth up to 255 m achieves the lowest collision rate of all, and in the other environments, our proposed method does the lowest collision rate. CycleGAN method [46] is higher collision rate than Shimada et al. method [34]. As shown in Table 1, CycleGAN [46] is higher accuracy than Shimada et al. [34]. According to these results, we can see that high-accuracy depth estimation alone does not reduce the collision rate and that the inference time needs to be shorter. The method using depth estimation would not reduce the collision rate more than the method dealing with ground truth. However, as the results show, the proposed method outperforms the method that handles ground truth. There are two reasons for these results. Firstly, the ground truth is an absolute distance, whereas the distance estimated by the depth estimation method is a relative distance (i.e. the maximum distance is always 255 m and the minimum distance is 0 m). The nearest obstacle in the depth image is always zero depth value in the depth estimation method so that the drone takes larger avoiding action than in the method dealing with the ground truth depth map. Therefore, the proposed method has the ability to record a lower collision rate than the ground truth. Finally, the depth estimation method sometimes estimates the depth of obstacles that are more than 255 m away. Therefore, in advance, the drone considers distant obstacles and determines waypoints has led to lower collision rates. From these results, we conclude that the proposed method provides sufficient estimation since it is able to select waypoints equivalent to those of the ground truth method despite the inference time.

In addition, there is a reason why our proposed method is superior to the other depth estimation methods [33,34]. As shown in Table 1, the proposed method is more accurate, and the estimation is performed many times during the flight. This iteration causes a situation where errors accumulate in the determination of waypoints. This accumulation of errors can lead to a fatal situation for the drone. Therefore, the proposed method results lower collision rate than [33,34] in all the environments.

## 5. Discussion

In this section, we will discuss what makes high accuracy with a short inference time. We also discuss the technical issues that need to be addressed.

First of all, we show how optical flow attention actually works and how it contributes. The proposed method is superior to the other methods because of the optical flow attention as shown in Tables 1, 2 and 5. Figure 7 shows visualization of optical flow attention in KITTI dataset.

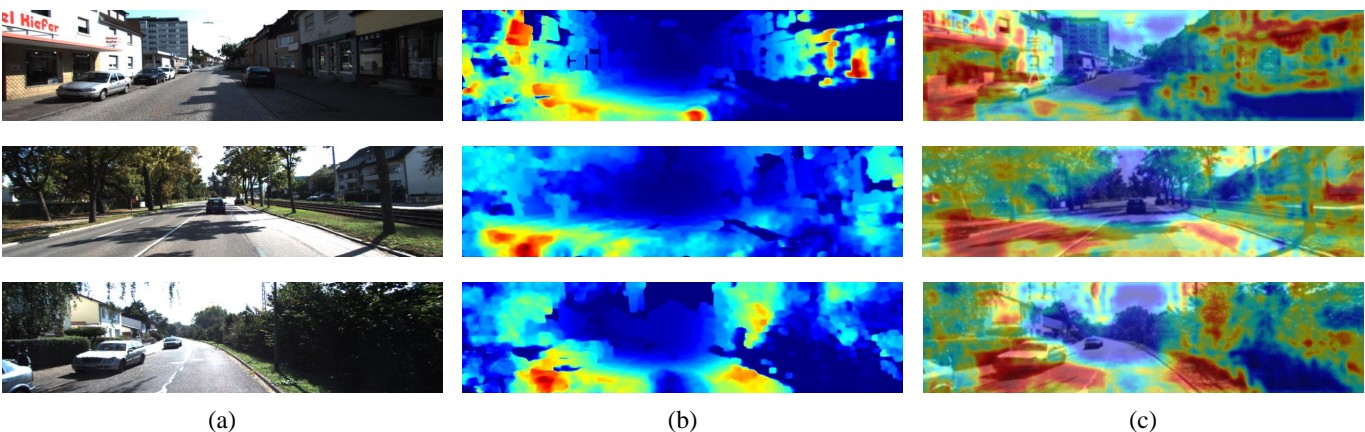

|  (a)  |  (b)  |  (c)  |

**Figure 7.** Visualization how the optical flow attention works in KITTI dataset: (**a**) monocular images, (**b**) optical flow maps, (**c**) visualization of attention.

As shown in Figure 7, the optical flow attention adds large attention to nearby objects. The optical flow attention can also add information to areas not represented by optical flow maps. The features of the monocular image enhanced by this attention are input to subsequent down-sampling, allowing more effective feature extraction than with monocular images. Therefore, the proposed method achieves high accuracy since the proposed method can utilize the optical flow information more than Shimada et al. method [34] which takes only a part of optical flow pixels.

In addition, as shown in Table 2, adding optical flow information by attention is possible to achieve the same accuracy as the Kuznietsov et al. method [18], which is a high-performance CNN. Figure 8 shows visual aspects using the KITTI dataset.

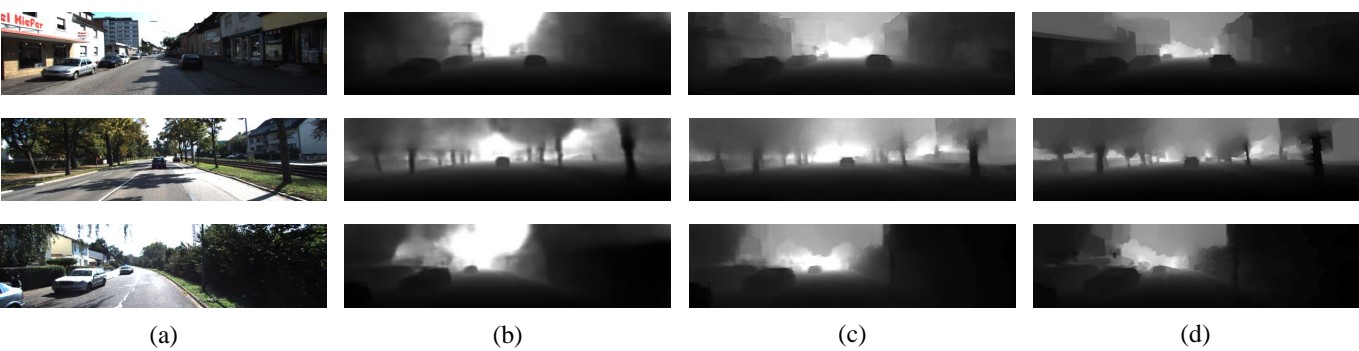

|  (a)  |  (b)  |  (c)  |  (d)  |

**Figure 8.** Visual aspects using KITTI dataset: (**a**) monocular images, (**b**) Kuznietsov et al. estimation results, (**c**) our results, (**d**) ground truth.

As shown in Figure 8, the proposed method has clear edges and the results are comparable to the Kuznietsov method [18]. On the other hand, the proposed method reflects the poles in the depth estimation results while the Kuznietsov method [18] does not show them in the depth estimation results. As shown in Figure 7c, the telephone poles have been highlighted by the optical flow attention with displacements detected in the previous and current frames. The shape of the car in side of a street is also clearer by the

optical flow attention than in the Kuznietsov method, and these differences are reflected in the accuracy of the proposed method. Therefore, without deepening the CNN, the optical flow attention improves the accuracy of depth estimation with fast inference time.

We describe the technical issues. Our proposed method employs an optical flow as a simplified depth. The simplified depth required by the optical flow requires the object to be moving between each frame. Therefore, optical flow acquired by a moving camera, such as a drone, can accurately calculate the simple depth outside the center of the image. On the other hand, in the central part of the image, there is very little or no movement between each frame of the moving camera and it does not show up in the optical flow. Objects moving at the same speed as the drone are similarly not represented in the optical flow. Figure 9 shows an example that optical flow does not work.

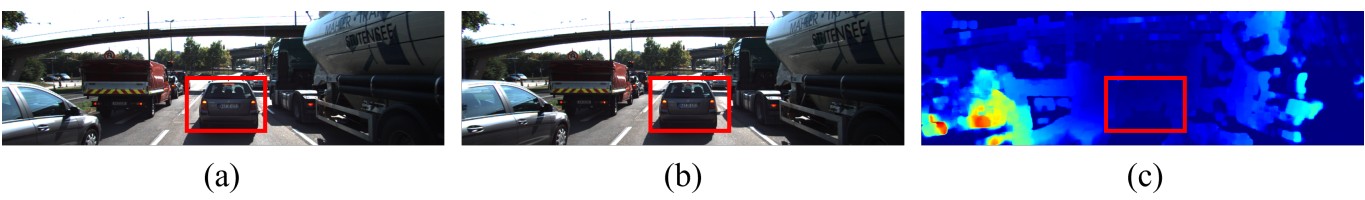

<div align="center">(a) (b) (c)</div>

**Figure 9.** An example the optical flow does not work: (**a**) previous frame, (**b**) current frame, (**c**) optical flow map.

As shown in Figure 9, the bounding box shows a car location however in the optical flow map, there is no information in the bounding box. Therefore there is a necessity to develop a more effective simplified depth generation method.

## 6. Conclusions

In this paper, we propose a fast inference time and high-accuracy depth estimation method for autonomous drones. To achieve a highly accurate depth estimation of monocular images alone, it is generally necessary to use a deeper CNN to extract more features. However, this is not suitable for drones due to the long inference time. We propose optical flow attention that does not deepen the network but rather inputs optical flow as information that can aid in-depth estimation. In addition, we add perceptual discriminators and skip connections to make our fast estimator more effective than the conventional training method.

Experimental results demonstrate our proposed method is superior to the state-of-the-art in accuracy, error, and collision rate with fast processing time. Our proposed method is considered to be the most suitable method for the autonomous flight of drones. We also conduct an ablation study to confirm the contributions of our proposed method. The results of the ablation study demonstrate that the skip connection contributes to edge sharpness, the optical flow attention contributes to accuracy improvement, and the perceptual discriminator contributes to preventing the vanishment of detailed object estimation.

In future work, we will investigate more efficient information than optical flow. In addition, as shown in Table 6, the collision rate in City Environment is too high to be feasible for real environments. Therefore, we will design more effective collision avoidance for depth estimation. There are also a number of challenges in the current technology of depth estimation for drones. These include, for example, reduced energy consumption, faster inference times, higher accuracy and fast and robust waypoint determination methods. Solving these problems is a future challenge for the field.

**Author Contributions:** Conceptualization, T.S.; Funding acquisition, X.K. and H.T.; Investigation, T.S.; Methodology, T.S.; Software, T.S.; Supervision, H.N., X.K. and H.T.; Validation, T.S.; Writing—original draft, T.S.; Writing—review and editing, H.N., X.K. and H.T. All authors have read and agreed to the published version of the manuscript.

**Funding:** This work is partly supported by JSPS KAKENHI Grant Numbers 20K23333 and 20H00590.

**Institutional Review Board Statement:** Not applicable.

**Informed Consent Statement:** Not applicable.

**Data Availability Statement:** Not applicable.

**Acknowledgments:** The authors appreciate all the institutions and individuals that have provided support for this paper.

**Conflicts of Interest:** The authors declare no conflict of interest.

## Abbreviations

The following abbreviations are used in this manuscript:

| | |
|---|---|
| CNN | Convolutional Neural Network |
| UAV | Unmanned Aerial Vehicles |
| VTOL | Vertical Take-Off and Land |
| HTOL | Horizontal Take-Off and Landing |
| DNN | Deep Neural Network |
| GPU | Graphical Processing Unit |
| CGAN | Conditional Generative Adversarial Network |
| SLAM | Simultaneous Localization and Mapping |
| VGG | Visual Geometry Group |

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
