# Peer review of "Fast and High-Quality Monocular Depth Estimation with Optical Flow for Autonomous Drones"

_drones, doi:10.3390/drones7020134_

Round 1
Reviewer 1 Report
Dear authors, thank you for an interresting article.
I have several questions and remarks:
1) abstract - if you yuse an abbreviation, which is not commonly known, define it
advanced deep CNNs
discriminator into CGAN
Both aren´t defined in the text.
2) introduction
my point of view - the introduction should be broader;
this should be expanded to include the typical use of drones with links. There is very little here, just one sentence (row 15-16). To top it off, multicopters and winged drones need to be defined. Most of the time it seems to be multicopters. But e.g. EBee or even other winged drones have an optical sensor (which sometimes fails in homogeneous surface).Add a brief list and links.
Determining the altitude is essential for a good landing! Put that in the text. See for example landing of Ebee drone.
Fallowing it is an example (suggestion):
agriculture
10.3390/s22197693
forestry
10.3390/rs14133205
military
10.13140/RG.2.2.25777.02402
mapping
10.1186/s40537-021-00436-8
http://dx.doi.org/10.5194/isprsarchives-XXXVIII-1-C22-277-2011
cadastre
http://dx.doi.org/10.5194/isprsarchives-XXXVIII-1-C22-57-2011
building documentation
10.1680/jmapl.19.00041
archaeology
10.1002/arp.1569
arctic reserch
10.3390/app11020754
etc.
Then you can move on to your research.
3) VGG row 80 define it
4) 3. Depth Estimation with Optical Flow
An important questions: how is it related with a) a SLAM technology? B) with stereophotogrammetry or SfM or MVS (multi view stereo)?
Write about it some sentences.
Your point of view is from the computer or mathematical side, photogrammetry solves the distance from two or more images for a very long time. Some reference and a few sentences would be ok.
Thank you.
Author Response
Dear Reviewer 1.
We also appreciate the time and effort you and each of Reviewer 1 has dedicated to providing insightful feedback on ways to strengthen our paper. Thus, it is with great pleasure that we resubmit our article for further consideration. We have incorporated changes that reflect the detailed suggestions you have graciously provided.
Please see the attachment.

Reviewer 2 Report
The authors shared with us a method to refine the depth estimation map for autonomous drones. Starting from generation of optical flow image by using monocular image, they added a perceptual discriminator into CGAN, as novelty of this paper, to improve accuracy of depth estimation and to estimate high-quality images in short time. They comapred their method with other depth estimation methods by using two kind of datasets, AirSim and KITTI. Technically, the paper shows lack of methodological points and results. Also, the layout of the paper not at all helpful for easy reading. The comments listed below will be helpful for authors to ameliorate their paper in order to be considered in high quality journal like ''Drones''.
I) Major comments :
1- It’s necessary to share with readers how much data you used in training and validation (or testing) for each dataset (AirSim and KITTI) to rain and validate the proposed model ?
2- Based on what you choose the paramters of the used model, including epoches number, batch size, α, β and γ for each dataset ?
3- Did the authors compared the results of their model with the results of the results of the other models in their published papers ? or they raning all models by themself and comapred the error and accuracy evaluation acheived from each model. If the second case, the authors are asked to share with us how they implemented the other model used in comparaision, their paramters, the training and validtation data used in each model.
4- The inferance time evaluation it’s ambiguous !!! As the authors shared in text (lines 254-261), they evaluate the inferance time of their model in comparaison with four other models by using AirSim datasets. The four models are Shimada et al. [11], CycleGAN [42], Xin et al. [35], and Kuznietsov et al. [9], the first two ones are already used in error and accuracy comaparision based on same dataset (AirSim), while the other ones were used in error and accuracy evaluation and comparaison based on KITTI datasets. Why these difference ? Its mendatory to share with us the inferance time evaluation of the other model based on corresponding used datatset in accuarcy and error evaluation comaparaison. Further, it’s necessary to add the inferance time evaluation for the rest models, e.g. Eigen et al.[19], Liu et al.[23], and Shimada et al.[10].
5- Ditto for ablation analyze, would be better to share with us the results the ablation study using KITTI datasets not only AirSim adatasets.
6- Ditto for collision rate, would be better to share with us the results for all methods from both datasets.
7- Absence of real discussion made the paper as comparative study ? So, would be better to add new section called discussion in wich its necessary to discuss your results, enrished by literature background and developed well.
II) Minor comments :
1- In its current form, the abstract, not at all informative and it’s like a conclusion!!! Would be better to share the evaluation metrics, infernance time, ablation and collision avoidance results in abstarct, hilighting the high performance of the proposed method. Also, would be better to hisghlight the novelty and the scientific contribution of the current study rather on similar ones in the literature.
2- Merge the two first sections, Introduction and Related work, avoidng redundauncy in text, and develop the introducation by up to date literature background.
3- Remove the shared text in lines 47-54 (Introduction section), it seems a conclusions of your study.
4- Add new secion called Methods, in which merge what you already furnished in section 3. Depth Estimation with Optical Flow, and what you described as methodological points in section 4. Experimental Results, e.g. 4.1. Implementation, 4.2. Accuracy, Error and Latency Evaluation, lines 254-261, 4.3. Ablation Study, and 4.4. Collision Rate Evaluation. Seperate the results from methods !!!
5- Figure 2 is part of what you shared in figure 1, also figure 1 is same to figure 3, except what you detailed in digure 3. So, would be better to remove figures 1 and 2, add new monocular image in figure 3 by adding new arrow indicating generation of optical flow map from both.
Overall, the paper needs too much and carefully work.
Best of luck
Author Response
Dear Reviewer 2.
We also appreciate the time and effort you and each of Reviewer 1 has dedicated to providing insightful feedback on ways to strengthen our paper. Thus, it is with great pleasure that we resubmit our article for further consideration. We have incorporated changes that reflect the detailed suggestions you have graciously provided.
Please see the attachment.

Round 2
Reviewer 2 Report
It’s great to see that the authors respond to all comments that reviewer raised during first revision round. They tried to incorporate major issues, but the paper still needs improvement to be considered for publication. Please take care of comments below :
- The abstract needs improvement, by minimizing the long sentences (first 7 lines) used for introduction, by including some results ; especialy error, accuracy, ablation, etc ; without forgeting to mention the comparaison, and by highlighting the scientific contribution of the current study.
- Mixing the results and experiments steps made the paper hard to follow, so please seperate the results from the section 3. Experiments, in new section called 4. Results.
- It's confusing to see that the authors discuss the importance of the optical attention in neighborhood case only avoiding the other cases (Blocks, City Environment, and Coastline), refer to discussion section. It's mondatory to add results of other sites.
- Redone the last sentence in conclusion ''In future work, we will investigate more efficient information than optical flow. In addition, as shown in Table 6, the collision rate in City Environment is too high to be feasible for real environments. Therefore, we will more effective collision avoidance for depth estimation.'' Would be better to highlight these remarks for all sceintific community not for yourself alone.
- Minor comments : check typo errors in text, e.g. line 235 replaced as train data by as training data, ...etc.
Best of luck.
Author Response
Dear Reviewer 2.
We also appreciate the time and effort you and each reviewer have dedicated to providing insightful feedback on ways to strengthen our paper. Thus, it is with great pleasure that we resubmit our article for further consideration. We have incorporated changes that reflect the detailed suggestions you have graciously provided.
Please see the attachment.
